# Cradle-to-Gate Water-Related Impacts on Production of Traditional Food Products in Malaysia

**P.X.H. Bong** [1,*], **M.A. Malek** [2], **N.H. Mardi** [3] **and Marlia M. Hanafiah** [4,5]

[1] Civil Engineering Department, College of Engineering, University Tenaga Nasional, Kajang 43000, Selangor, Malaysia

[2] Institute of Sustainable Energy, University Tenaga Nasional, Kajang 43000, Selangor, Malaysia; marlinda@uniten.edu.my

[3] Department of Civil Engineering, University Tenaga Nasional, Kajang 43000, Selangor, Malaysia; NHani@uniten.edu.my

[4] Department of Earth Sciences and Environment, Faculty of Science and Technology, University Kebangsaan Malaysia, Bangi 43600, UKM, Selangor, Malaysia; mhmarlia@ukm.edu.my

[5] Center for Tropical Climate Change System, Institute of Climate Change, University Kebangsaan Malaysia, Bangi 43600, UKM, Selangor, Malaysia

[*] Correspondence: SC22682@utn.edu.my

**Abstract:** Modern technology and life-style advancements have increased the demand for clean water. Based on this trend it is expected that our water resources will be under stress leading to a high probability of scarcity. This study aims to evaluate the environmental impacts of selected traditional food manufacturing products namely: tempe, lemang, noodle laksam, fish crackers and salted fish in Malaysia. The cradle-to-gate approach on water footprint assessment (WFA) of these selected traditional food products was carried out using Water Footprint Network (WFN) and Life Cycle Assessment (LCA). Freshwater eutrophication (FEP), marine eutrophication (MEP), freshwater ecotoxicity (FETP), marine ecotoxicity (METP) and water consumption (WCP), LCA were investigated using ReCiPe 2016 methodology. Water footprint accounting of blue water footprint ($WF_{blue}$), green water footprint ($WF_{green}$) and grey water footprint ($WF_{grey}$) were established in this study. It was found that total water footprint for lemang production was highest at 3862.13 $m^3$/ton. The lowest total water footprint was found to be fish cracker production at 135.88 $m^3$/ton. Blue water scarcity ($WS_{blue}$) and water pollution level (WPL) of these selected food products were also determined to identify the environmental hotspots. Results in this study showed that the $WS_{blue}$ and WPL of these selected food products did not exceed 1%, which is considered sustainable. Based on midpoint approach adopted in this study, the characterization factors for FEP, MEP, FETP, METP and WCP on these selected food products were evaluated. It is recommended that alternative ingredients or product processes be designed in order to produce more sustainable lemang.

**Keywords:** life cycle assessment; water footprint; impact assessment

---

## 1. Introduction

Currently, water scarcity is one of the key issues worldwide. A large amount of water is used and polluted by human activities. Worldwide, large volumes of water are consumed in agriculture, industry and domestically along with their respective pollutants [1,2]. The desire to decrease carbon footprint is routinely recognized, but the urgency to decrease water footprint (WF) is widely dismissed.

The relationship between environment and industry is essential for long-term industrial performance. The processes have placed increasing burdens on the environment through water pollution and waste products. The food industries are responsible for significant environmental

impacts but they contribute to human health and prosperity. The food market is concerned with food sustainability with the growing number of consumers demanding licensed sustainable agricultural products. According to Barsato et al. (2019) [3], total WF for a 0.75 L bottle of Italian wine is 1.193 m$^3$ while the value of freshwater eutrophication is $4.82 \times 10^{-3}$ kg P eq. Based on Miguel et al. (2015) [4], the WF of Spanish pork production averages 19.5 billion m$^3$/yr over the period of 2001–08. Results of the study established by Muhammad-Muaz and Marlia (2015) showed that total WF for growing oil palm is 243 m$^3$/ton [5]. These suggest that there is a need to reduce WF and its impact to the environment of food products.

As a developing country, it is necessary to revise, evaluate and mitigate the effects of current water use for future generations to safeguard the quantity and quality of water resources in Malaysia. To this end, Hoekstra established a concept of WF in 2002 [6] through Water Footprint Assessment (WFA) approach. Water footprint (WF) offers a comprehensive framework to evaluate enhancement possibilities for a specified industrial scheme by quantifying more than just direct water use [7]. Water footprint is a holistic tool that measures direct and indirect water use and pollution. The component of indirect water use in WF is connected to the concept of virtual water [8]. It accounts for water embodied in a product production chain through cumulative consumption. In mapping the magnitude of water use, WF highlights the importance of looking at the full supply chain of products.

Water Footprint Assessment studies in Malaysia were mostly conducted by the Malaysian Palm Oil Board (MPOB) and the Standard and Industrial Research Institute of Malaysia (SIRIM Berhad) [9]. Nevertheless, WFA studies conducted on food production in Malaysia are scarce. Furthermore, despite various green policies, no regulations were issued by the Malaysia government regarding food production. For the purpose of sustainability, it is essential to evaluate the environmental performance of food production in Malaysia using appropriate guidelines such as conceptual framework established by WFA. This would help the Malaysia government and associated stakeholders to make informed decisions to enhance the environmental performance of food products.

Assessments of the feasibility of food production using WFA have yet to be established in Malaysia. This study therefore seeks to fill this gap. In this study, a cradle-to-gate approach was adopted from the extraction of raw materials used to the food production line as observed at the factory. WFA was then determined and several environmental impacts to water were investigated in this study. This study aims to quantify WF using WFN approach and assess the environmental impacts of food production using an LCA approach that is fully ISO standard compliant. Initiatives conducted in this study would improve the efficiency and sustainability of these food products.

## 2. Materials and Methods

### 2.1. Goal and Scope Definition

The system boundary is schematically presented in Figure 1. In this study, the system boundary is cradle-to-gate, i.e., from cultivation or raw materials to grow the ingredients in the selected food products until the completion of the food product before leaving the factory. In addition, the energy and material requirements as well as the emissions into the environment were taken into consideration. Retail, shipping and final consumption phases of these food products are not included in the scope of this study.

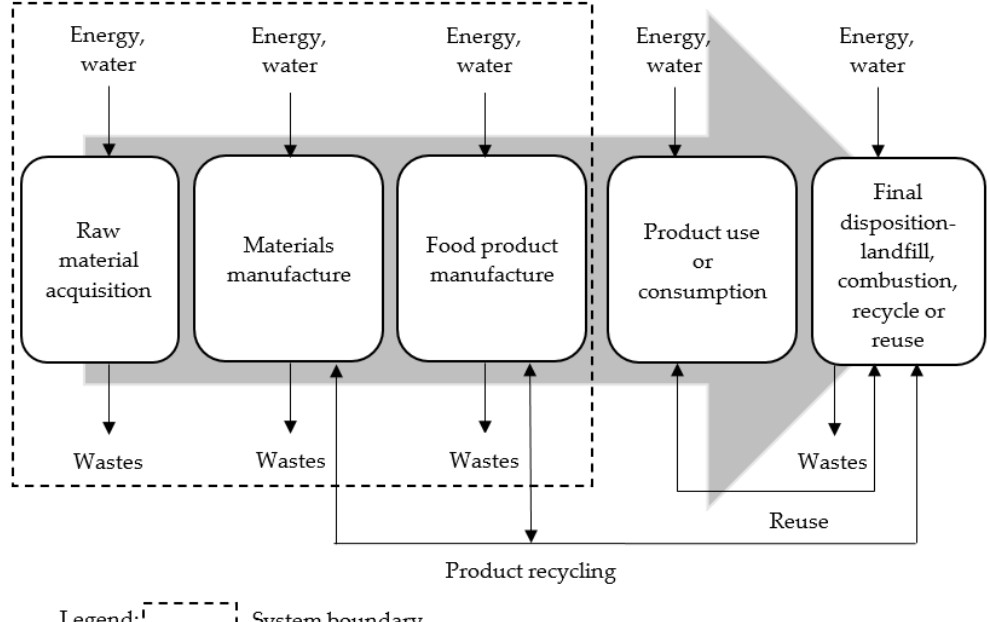

**Figure 1.** System boundary of Water Footprint Assessment and Life Cycle Assessment for the production of 5 traditional food products: tempe, lemang, noodle laksam, fish cracker and salted fish.

Description of Food Products

In this study, five Malaysia traditional food products were analyzed: tempe (TP), lemang (LM), noodle laksam (LS), fish cracker (FC) and salted fish (SF). These traditional food products were selected because they can be found in every state in Malaysia and the per capita consumption of these food products was high. According to a survey made by Ministry of Health Malaysia, the consumption of TP, LM, LS, FC and SF were 30,330 ton/yr, 21,956 ton/yr, 91,927 ton/yr, 13,455 ton/yr and 4837 ton/yr respectively [10].

Tempe originated in Indonesia and is made primarily from soybeans [11]. It is made through natural culturing and a controlled fermentation process that binds the soybeans to produce higher contents of protein, vitamins, and dietary fiber. The ingredients to make TP are soybean and yeast. Lemang is made from glutinous rice, coconut milk, salt, water, banana leave and bamboo sticks. It is made through a soaking and cooking process that places a soaked glutinous rice mixture with coconut milk and salt inside bamboo sticks lined with banana leaves. Noodle laksam is a type of noodle which originated from the states of Kelantan and Terengganu in East Malaysia. The process to produce LS is steaming the mixtures by mixing all the ingredients which are rice flour, wheat flour and water. Fish crackers are a deep fried cracker commonly found in Malaysia. They are made through a mixing and boiling process. The ingredients are fish, starch, egg, ice, salt, seasoning, sugar and water. Salted fish is a staple diet for the Malaysians that provides preserved animal protein. The process to produce SF is salting and drying the fish.

*2.2. Water Footprint Accounting*

Currently, there are two main approaches established in WFA [11,12]. One of these approaches was established by Water Footprint Network (WFN), an organization that defines WF as an indicator of freshwater consumption, not only direct use of water by consumers or producers, but also for indirect use of water [13]. Another WF approach was created by Water Use in Life Cycle Assessment (WULCA) research group. It established the water scarcity midpoint method for Life Cycle Assessment (LCA) and assessments of water scarcity footprint [14]. This LCA-based WFA involves quantification of water impacts related to freshwater consumption in terms of water availability footprint and water scarcity footprint. The International Standard Organization (ISO) published this approach in

2014 as an international standard entitled "Environmental Management-Water Footprint - Principles, Requirements and Guidelines" [15].

The WF of a product is defined as the total volume of freshwater used to produce a product, measured by considering the entire product supply chain [13]. Water Footprint Assessment is an analytical approach that provides new perspective to develop an adaptable technique of water management for products, processes, or organizations [16,17].

Water footprint is divided into three components, namely blue WF, green WF and grey WF. Blue WF ($WF_{blue}$) is a measure of consumptive use of freshwater or groundwater to produce a product. Green WF ($WF_{green}$) is the volume of rainwater consumed during the production process which is mostly related to agricultural and forestry products. It refers to total rainwater evapotranspiration and water used in a harvested crop or wood. Grey WF ($WF_{grey}$) is defined as the volume of freshwater required to dilute the load of pollutants in a water body to meet existing ambient water quality standards [18].

Data to be used in the accounting of WF were collected on site at the identified study areas. All the data were obtained from food factories located in Peninsular Malaysia. The most important element in this investigation is the food production process flow line in the factories which leads to the knowledge of water used in each process for WF accounting.

For accounting of $WF_{blue}$, blue water is incorporated and extracted from water bills at the identified factories, while the lost return flow was obtained from the production process flow. Values on production quantity, P[p] and fraction of output product, $f_v$[p] were also obtained from the identified factories. For $WF_{grey}$, wastewater samples were collected on-site, and various water quality tests were then conducted at the laboratory. However, $WF_{green}$ for food production in factory is not considered in this study since none of the identified factories used rainwater during their production process. $WF_{green}$ is mostly related to agricultural and forestry products, where total rainwater and evapotranspiration are incorporated into the harvested crops or wood [19]. Hence, only $WF_{green}$ for raw materials production is considered in this study. The functional unit is defined as $m^3$ per ton of each food product.

Table 1 shows the references for major ingredients and materials used in selected food products. It should be noted that limited data provided by factory owners is due to market competition among food producers in the country. The values of WF in raw materials for the selected food products were adopted from Mekonnen and Hoekstra (2010) [20,21], Kandananond and Author (2018) [22], Water Footprint Network (2007) [23], Franzén (2014) [24] and Ecoinvent 3.4 Database [25].

**Table 1.** References of water footprint (WF) for raw materials used for 5 traditional Malaysian foods.

| Food Product | Materials | Reference |
|---|---|---|
| Tempe (TP) | Soybean | Mekonnen, Hoekstra, 2010b [21] |
| | Yeast | Franzén, 2014 [24] |
| | Liquefied petroleum gas (LPG) | Kandananond, Author, 2018 [22] |
| | Electricity | Kandananond, Author, 2018 [22] |
| | Plastic | Water Footprint Network, 2007 [23] |
| Lemang (LM) | Glutinous rice | Mekonnen, Hoekstra, 2010b [21] |
| | Coconut milk | Mekonnen, Hoekstra, 2010b [21] |
| | Salt | Mekonnen, Hoekstra, 2010b [21] |
| Noodle laksam (LS) | Rice flour | Mekonnen, Hoekstra, 2010b [21] |
| | Wheat flour | Mekonnen, Hoekstra, 2010b [21] |
| | LPG | Kandananond, Author, 2018 [22] |
| | Plastic | Water Footprint Network, 2007 [23] |

**Table 1.** *Cont.*

| Food Product | Materials | Reference |
|---|---|---|
| Fish cracker (FC) | Fish | Ecoinvent 3.4 Database [25] |
| | Starch | Ecoinvent 3.4 Database [25] |
| | Egg | Mekonnen, Hoekstra, 2010a [20] |
| | Salt | Mekonnen, Hoekstra, 2010b [21] |
| | Sugar | Mekonnen, Hoekstra, 2010b [21] |
| | Plastic | Water Footprint Network, 2007 [23] |
| | Electricity | Kandananond, Author, 2018 [22] |
| Salted fish (SF) | Fish | Ecoinvent 3.4 Database [25] |
| | Salt | Mekonnen, Hoekstra, 2010b [21] |
| | Plastic | Water Footprint Network, 2007 [23] |

### 2.3. Water Footprint Sustainability Assessment

The Water Footprint Assessment Manual provides a clear guideline for the accounting approach of the assessment [18]. In order to have an indication of the severity at a hotspot, blue water scarcity ($WS_{blue}$) and water pollution level (WPL) are required to be calculated. When the environmental water needs are exceeded or when pollution exceeds the waste assimilation capacity, WF in that catchment area is environmentally unsustainable [18].

Blue water scarcity in a catchment x is defined as the ratio of total blue water footprints in the catchment ($\Sigma WF_{blue}$) to blue water availability ($WA_{blue}$):

$$WS_{blue}[x, t] = \frac{\sum WF_{blue}[x, t]}{WA_{blue}[x, t]} \tag{1}$$

If the $WF_{blue}$ exceeds the $WA_{blue}$, this implies that the $WF_{blue}$ is environmentally unsustainable and environmental flow requirement (EFR) is violated. A $WS_{blue}$ of 100 per cent means that the available blue water has been fully consumed. A blue water scarcity beyond 100 percent is not sustainable.

The effect of total $WF_{grey}$ in a catchment depends on runoff in the catchment available to assimilate the waste. $WF_{grey}$ in a specific period at a specific catchment forms a hotspot when ambient water quality standards of that period in that catchment are violated, in other words, when waste assimilation capacity is fully consumed.

As a relevant local impact indicator, WPL within a catchment, which measures the degree of pollution, can be calculated. It is defined as fraction of waste assimilation capacity consumed. It is calculated by taking the ratio of total $WF_{grey}$ in a catchment ($\Sigma WF_{grey}$) to the actual runoff from that particular catchment ($R_{act}$). Water pollution level of 100% means that the waste assimilation capacity has been fully consumed. When WPL is above 100%, ambient water quality standards are violated. WPL is thus calculated for a catchment at a particular time as follows:

$$WPL[x, t] = \frac{\sum WF_{grey}[x, t]}{R_{act}[x, t]} \tag{2}$$

### 2.4. Water-Related Impact Assessment

This study addresses water-related impact issues related to the Malaysia traditional food products. Life Cycle Assessment is a commonly used method to assess a product, process or environmental impacts resulting from these services throughout its life cycle [22–24]. It is also an appropriate method to analyze the environmental performance of the food industry [25,26]. Life Cycle Assessment framework

consists of four phases, which are goal and scope definition, inventory analysis, environmental impact assessment and interpretation of results [27,28].

### 2.4.1. Goal and Scope Definition

The goal and scope phase is the first phase in an LCA study. Food production and manufacturing are increasing due to fast population growth and changing lifestyles, resulting in greater consumption of global resources [29]. These industries utilize large amounts of raw materials, energy and water consumption, in addition to the disposal of waste in the environment, causing major negative impacts on the environment. The goal of this study was to identify the water-related impacts of 5 traditional food products in Malaysia. The scope was chosen based on the goal. The functional unit is represented by 1 ton of each food product.

A few published LCA studies in food production have been reviewed as shown in Table 2. The studies compare food types, functional unit, system boundary and impact. Most environmental impact studies for food products have focused on ecotoxicity potential, acidification potential and eutrophication potential.

**Table 2.** General methodology of life cycle assessment (LCA) studies related to food products.

| Authors, Year | Food Types | Functional Unit | System Boundary | Method | Impact Categories |
|---|---|---|---|---|---|
| Lansche et al., (2020) [30] | Crisps | 1 kg of crisps | Cradle-to-gate | SALCA | CED, DEF, WSI, GWP, OFP, AP, HTP, ETP |
| Borsato et al., (2019) [3] | Wine | 750 mL of wine | Cradle-to-gate | AWARE, IMPACT 2002+, ILCD 2001, ReCiPe 2016 | WSI, AP, ETP, EP |
| Bhattacharyya et al., (2019) [31] | Vodka | 750 mL of vodka | Cradle-to-grave | ReCiPe 2016 | ETP, EP, CC, LU |
| Perez-Martinez et al., (2018) [32] | Pork lean can, meatballs with peas can | 220 g of pork lean can, 430 g of meatballs with peas can | Cradle-to-grave | ReCiPe 2016 | ETP, CC, DEP, AP, EP, LU |
| Caldeira et al., (2018) [33] | Palm rapeseed, soya, waste cooking oil | 1 kg of each food product | Cradle-to-gate | IMPACT 2002+, ReCiPe 2008, USETox | ETP, EP, AP |
| Jimmy et al., (2017) [34] | Rice | 1 kg of rice | Gate-to-gate | ReCiPe 2016 | ETP, CC, DEP, AP, EP, LU |
| Cha et al., (2017) [35] | White radish | 1 ton of white radish | Cradle-to-gate | EUTREND, ReCiPe 2016 | EP, WD |
| Huerta et al., (2016) [36] | Boneless beef, fatless beef | 1 kg of boneless beef, 1 kg of fatless beef | Cradle-to-grave | ReCiPe 2008 | ETP, CC, DEP, AP, EP, LU |
| Nilsson et al., (2011) [37] | Crisps | 200 g of crisps | Cradle-to-gate | IPCC, CML2001 | GHG, EP, EU |
| Bevilacqua et al., (2007) [38] | Pasta | 0.5 kg of pasta | Cradle-to-grave | EDIP, EPS 2000, CML 2, Eco-indicator 99 | ETP, AP, EP, LU |

Notes: ecotoxicity potential (ETP), acidification potential (AP), eutrophication potential (EP), land use (LU), water degradation (WD), water degradation footprint (WDF), water eutrophication footprint (WEF), water stress index (WSI), global warming potential (GWP), ozone formation potential (OFP), human toxicity potential (HTP), greenhouse emissions (GHG), energy use (EU), climate change (CC), depletion (DEP), cumulated energy demand (CED), deforestation (DEF).

### 2.4.2. Life Cycle Inventory (LCI)

In this study, the LCI includes inventory data from the identified inputs and desired outputs in the process flow of producing the selected traditional food products. The inventory data consist of the amount of water, energy and materials consumed and the quantities of emissions released to the water [39]. The preparation of inventory data included primary and secondary data. The primary data consisted of questionnaire and laboratory results obtained from water quality samples collected

at identified factories. Meanwhile, the secondary data were obtained from companies, authorities, literature review and selected databases.

Table 3 summarizes the inventory data used for all selected food products in this study. Column 1 is types of 1 ton of products. The elements in Column 2 were obtained from on-site questionnaires and on-site evaluation at identified food factories and literature reviews. The data in Column 3 were obtained from companies, authorities, literature reviews and selected databases. The ingredients were calculated based on the usage per ton of product. The amount of emissions produced during the acquiring and manufacturing processes of ingredients listed in Column 3 were provided by the selected databases, namely Ecoinvent 3.4 [20] and Agri-footprint 4.0 [40]. The quantities in Column 4 were obtained from data collection in Column 3 and calculated based on the functional unit of 1 ton of each food product.

**Table 3.** Inventory data of various ingredients for 5 traditional Malaysian foods.

| 1 Ton of Products (1) | Parameter Input/Output (2) | Ingredient (3) | Quantity (4) |
|---|---|---|---|
| Tempe (TP) | Resource | Water | 1491.228 L |
| | Materials/fuels | Soybean | 0.292 ton |
| | | Liquefied petroleum gas (LPG) | 4.094 kg |
| | | Plastic | 17.5000 kg |
| | Electricity/heat | Electricity | 10.526 kWh |
| | Emissions to water | Total suspended solids (TSS) | 3.395 mg |
| | | Chemical oxygen demand (COD) | 700,526.376 mg |
| | | Biological oxygen demand (BOD) | 33,115.792 mg |
| | | Ammonia | 390,065.823 mg |
| Lemang (LM) | Resource | Water | 81.019 L |
| | Materials/fuels | Glutinous rice | 833.333 kg |
| | | Coconut milk | 833.333 kg |
| | | Salt | 33.333 kg |
| | Emissions to water | TSS | 3.188 mg |
| | | COD | 2,470,671.294 mg |
| | | BOD | 62,165.277 mg |
| | | Ammonia | 12,353.356 mg |
| Noodle laksam (LS) | Resource | Water | 1402.778 L |
| | Materials/fuels | Wheat flour | 222.222 kg |
| | | Rice flour | 444.444 kg |
| | | LPG | 124.444 kg |
| | | Plastic | 100.000 kg |
| | Emissions to water | TSS | 15.980 mg |
| | | COD | 1,764,458.333 mg |
| | | BOD | 67,360.139 mg |
| | | Ammonia | 41,614.583 mg |

**Table 3.** *Cont.*

| 1 Ton of Products (1) | Parameter Input/Output (2) | Ingredient (3) | Quantity (4) |
|---|---|---|---|
| Fish cracker (FC) | Resource | Water | 200.000 L |
| | Materials/fuels | Fish | 1000.000 kg |
| | | Salt | 1127.300 kg |
| | | Sugar | 8.500 kg |
| | | Starch | 1100.000 kg |
| | | Egg | 100 p |
| | | Plastic | 17.500 kg |
| | Electricity/heat | Electricity | 340.000 kWh |
| | Emissions to water | TSS | 13.922 mg |
| | | COD | 2,241,066.667 mg |
| | | BOD | 36,926.667 mg |
| | | Ammonia | 444,096.222 mg |
| Salted fish (SF) | Resource | Water | 829.034 L |
| | Materials/fuels | Fish | 2000.000 kg |
| | | Salt | 555.556 kg |
| | | Plastic | 46.667 kg |
| | Emissions to water | TSS | 74.545 mg |
| | | COD | 3,313,848.214 mg |
| | | BOD | 62,753.036 mg |
| | | Ammonia | 656,467.232 mg |

Based on on-site data, it was found that 1491.228 L of water were used to produce 1 ton of TP in the factory. In terms of materials used, it was found that the highest ingredient was soybean at 0.292 ton and the lowest was LPG at 4.094 kg. The amount of electricity at 10.526 kWh was obtained from electricity bills at the factory. The value of TSS, COD, $BOD_5$ and Ammonia were acquired from the water quality tests on wastewater samples performed in the laboratory. The wastewater samples were collected from pipes that discharged to the drainage of the identified factory. Based on results obtained from laboratory tests conducted in this study, it was found that the production of churro bowl (CB) resulted to the highest value of COD at 0.7 kg and the lowest value of TSS at 3.395 mg.

Based on Table 3, it was found that 81.019 L of water were used to produce 1 ton of LM in the factory. The highest ingredient in terms of materials used was glutinous rice and coconut milk with both at 833.333 kg, and the lowest was salt at 33.333 kg. Based on on-site investigation, it was found that there is no electricity used in the production of LM in the factory. Results from laboratory tests carried out in this study show that LM production resulted in the highest COD value of 2.47 kg and the lowest TSS value at 3.188 mg.

Based on on-site data, it was found that 1402.778 L of water were used to produce 1 ton of LS in the factory. In terms of materials used, it was found that the highest ingredient was rice flour at 444.444 kg and the lowest was plastic at 100 kg. Based on on-site investigation, it was found that there is no electricity used in the production of LS in the factory. Based on results obtained from laboratory tests conducted in this study, it was found that the production of LM resulted to the highest value of COD at 1.764 kg and the lowest value of TSS at 15.98 mg.

Based on Table 3, it was found that 200 L of water were used to produce 1 ton of FC in the factory. The highest ingredient in terms of materials used was salt at 1,127.3 kg and the lowest was sugar at 8.5 kg. The amount of electricity at 340 kWh was obtained from electricity bills at the factory. Results from laboratory tests carried out in this study show that FC production resulted in the highest COD value of 2.241 kg and the lowest TSS value at 13.922 mg.

Based on Table 3, it was found that 829.034 L of water were used to produce 1 ton of SF in the factory. The highest ingredient in terms of materials used was fish at 2 tons and the lowest was plastic used for packaging at 46.667 kg. Based on on-site investigation, it was found that there is no electricity used in the production of SF in the factory. Results from laboratory tests carried out in this study show that SF production resulted in the highest COD value of 3.314 kg and the lowest TSS value at 74.545 mg.

### 2.4.3. Life Cycle Impact Analysis (LCIA)

The impact category indicators were calculated using ReCiPe 2016, developed by Huijbregts et al. (2016) [41] at the midpoint level. The potential impacts were determined using results obtained from inventory analysis. The specific midpoint characterization factor for the identified five damage impact categories were discussed. The five damage impact categories were freshwater eutrophication (FEP), marine eutrophication (MEP), freshwater ecotoxicity (FETP), marine ecotoxicity (METP) and water consumption (WCP).

Freshwater eutrophication potential (FEP) is expressed in kg Phosphorus equivalent (kg P eq). Marine eutrophication potential (MEP) was expressed in kg Nitrogen equivalent (kg N eq). Ecological toxicity potential refers to the freshwater and marine ecotoxicity. Both were expressed in kg 1, 4-dichlorobenzene (kg 1, 4-DCB). Water consumption (WCP) corresponds to amount of water that the watershed of origin is losing. The characterization factor for WCP is $m^3$ of water consumed per $m^3$ of water extracted.

## 3. Results and Discussion

### 3.1. Water Footprint Accounting

Figure 2 summarizes the amount of $WF_{blue}$, $WF_{green}$ and $WF_{grey}$ in each selected traditional food type. It should be noted that the limited data provided by the factory owner is due to market competition among food producers in the country. Values of WF in raw materials for these selected food products were adopted from References [21–24] and Ecoinvent 3.4 Database [25].

Based on Figure 2, it was found that the highest $WF_{blue}$ is consumed by SF at 382.281 $m^3$/ton because blue water is consumed in the production of salt. $WF_{blue}$ for salt is 368.939 $m^3$/ton. LS for $WF_{blue}$ was the second highest at 334.964 $m^3$/ton. This is due to the amount of $WF_{blue}$ used to produce rice flour and wheat flour. This study has found that the least significant contribution to $WF_{blue}$ was TP at 25.473 $m^3$/ton. The main contribution to $WF_{blue}$ in TP was during soybean production at 20.468 $m^3$/ton. FC for $WF_{blue}$ at 41.651 $m^3$/ton was the second lowest. The $WF_{blue}$ in FC was mainly from blue water consumption in starch and salt production.

In this study, $WF_{green}$ was calculated based on green water consumed for the ingredients used to produce the selected food products. Based on site investigation it was found that no green water was used to prepare the selected food products at the respective factories, but was used in the raw materials. Based on Figure 2, $WF_{green}$ in LM was the highest as compared to other food products at 3218.634 $m^3$/ton. Green water consumption was highest in coconut milk used in LM at 2224.167 $m^3$/ton. The second highest $WF_{green}$ was used in LS at 1087.111 $m^3$/ton. The $WF_{green}$ in rice flour was 800 $m^3$/ton and is the main contributor to $WF_{green}$ in LS. $WF_{green}$ in FC was found to be the lowest at 52.755 $m^3$/ton. Whereas in $WF_{green}$ salt was found to be the most significant contributor. In addition, the second lowest $WF_{green}$ was used in TP at 595.614 $m^3$/ton. The only contribution to $WF_{green}$ in TP was the green water consumption of soybean production.

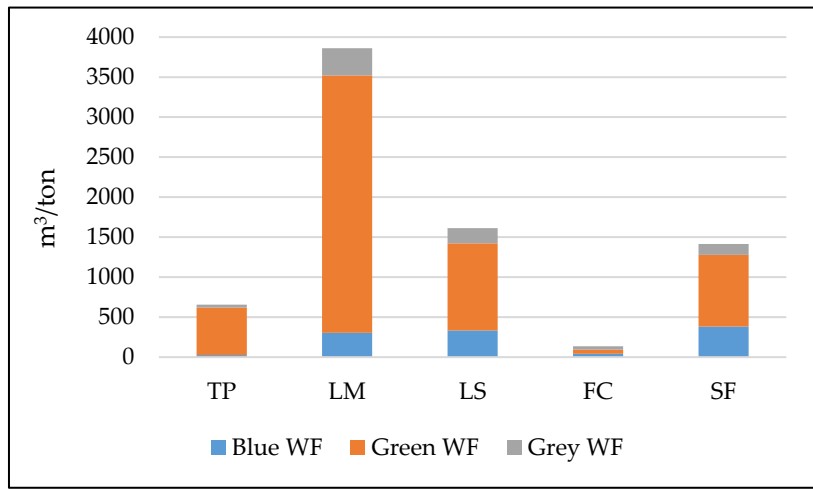

**Figure 2.** Calculated blue water footprint (Blue WF), green water footprint (Green WF) and grey water footprint (Grey WF) in m$^3$/ton of selected traditional food products from cradle-to-gate. TP is tempe, LM is lemang, LS is noodle laksam, FC is fish cracker and SF is salted fish.

WF$_{grey}$ was calculated based on water consumed to assimilate the pollutants discharged from respective factories with reference to standards of Environmental Quality (Industrial Effluent) Regulations 2009, Environmental Quality Act (EQA) 1974 (Act 127) [42]. In this study, WF$_{grey}$ in ingredients used were incorporated into the calculation. Based on Figure 2, it was found that the highest WF$_{grey}$ was produced by LM at 340.46 m$^3$/ton because freshwater was used to treat wastewater for LM production at the factory and for production of glutinous rice were 167.594 m$^3$/ton and 155.833 m$^3$/ton respectively. WF$_{grey}$ of LS was found to be the second highest at 189.866 m$^3$/ton due to the production of rice flour. The lowest contribution to WF$_{grey}$ was TP at 34.64 m$^3$/ton, of which blue water consumption during TP production in the factory was 23.751 m$^3$/ton. WF$_{grey}$ of FC at 41.475 m$^3$/ton was the second lowest, mainly from blue water consumption in FC production at the factory.

Based on findings from this study, it was found that LM has the highest total WF at 3862.133 m$^3$/ton as compared to others. The most significant contribution to LM is its WF$_{green}$. Based on Mekonnen and Hoekstra (2010) [21], similar product of LM, the WF$_{green}$ of cooked rice production also was the highest compared to its WF$_{blue}$ and WF$_{grey}$. It showed that large amounts of green water were consumed during the cultivation process.

*3.2. Scarcity Potential*

Table 4 summarizes values of WS$_{blue}$ and WPL for the selected traditional food products in this study. Based on Table 4, all the selected food products in this study were found to be sustainable since WS$_{blue}$ and WPL values do not exceeded 1.0. The highest WS$_{blue}$ and WPL value was found to be LS and the lowest was TP.

**Table 4.** Value for blue water scarcity (WS$_{blue}$) and water pollution level (WPL) for selected traditional food products in Malaysia.

| Food Products | WS$_{blue}$ | WPL |
|---|---|---|
| Tempe (TP) | 0.000045 | 0.000037 |
| Lemang (LM) | 0.000533 | 0.000359 |
| Noodle laksam (LS) | 0.002715 | 0.000923 |
| Fish cracker (FC) | 0.000070 | 0.000042 |
| Salted fish (SF) | 0.001394 | 0.000294 |

The method of calculating the WF of selected food products in terms of consumed or degraded water volumes is consistent with the WFN approach used by Mekonnen and Hoekstra (2012) [43]. The WF$_{grey}$ assessment in this study was limited to COD, BOD, TSS and ammonia nitrogen. The inclusion of phosphorus in the WF$_{grey}$ assessment could cover issues of aquatic ecotoxicity or acidification to complete the water quality impact assessment [10,38]. The use of the WPL at river basin scale allows the volumetric approach towards an indicator of degradative water use [38].

### 3.3. Life Cycle Analysis Results

In this study, the effects of the selected food to five damage impact categories were investigated. The five damage impact categories are FEP, MEP, FETP, METP and WCP. These five damage impact categories were selected in order to interpret the level (%) of detrimental effects to water. Figures 3–7 show the percentage contributions of five selected food products to the identified five damage impact categories. Results obtained in this study found that, compared to other selected food products, the production of FC was the main contributor to FEP and METP. On the other hand, the production of LM was found to be the main contributor to MEP and WCP. The production of LS was found to contribute the highest percentage of FETP and contributed the least to FEP. The production of TP was also found to contribute the lowest percentage to FETP, METP and WCP while SF production contributed the least to MEP.

Figure 3 shows that the soybean used in TP production represents the greatest contribution to FEP (95%), MEP (91%), FETP (93%) and METP (83%), while TP production process in the factory was responsible for the highest contribution to WCP (74%). As observed in the results (Figure 4), the glutinous rice used in LM production caused the highest contribution to FEP, MEP, FETP, METP and WCP at 77%, 62%, 81%, 93% and 99% respectively. Based on Figure 5, the usage of rice flour from LS production contributed the highest percentage to FEP (52%), MEP (67%), FETP (94%), METP (88%) and WCP (96%). Figure 6 shows that the starch used from FC production was responsible for the highest contribution to FEP, MEP, FETP, METP and WCP at 61%, 93%, 67%, 70% and 57% respectively. From Figure 7, the results show that the salt used in SF production represents the greatest contribution to FEP (88%), MEP (92%), FETP (88%), METP (88%) and WCP (62%).

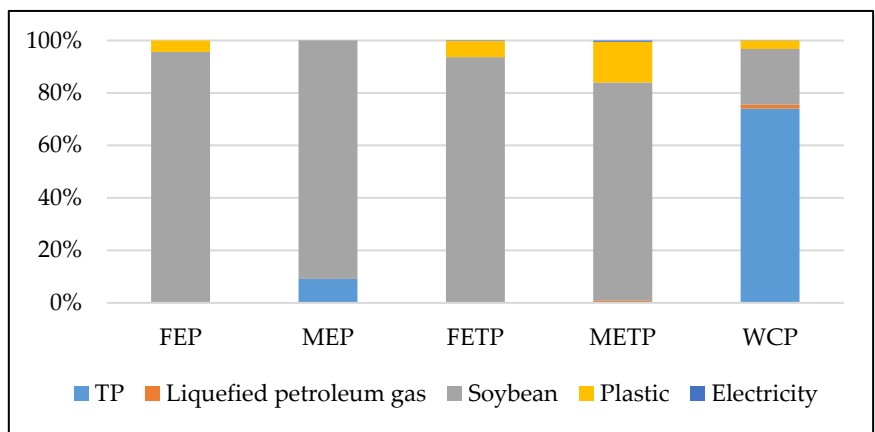

**Figure 3.** Relative contribution of tempe (TP) production to freshwater eutrophication (FEP), marine eutrophication (MEP), freshwater ecotoxicity (FETP), marine ecotoxicity (METP) and water consumption (WCP).

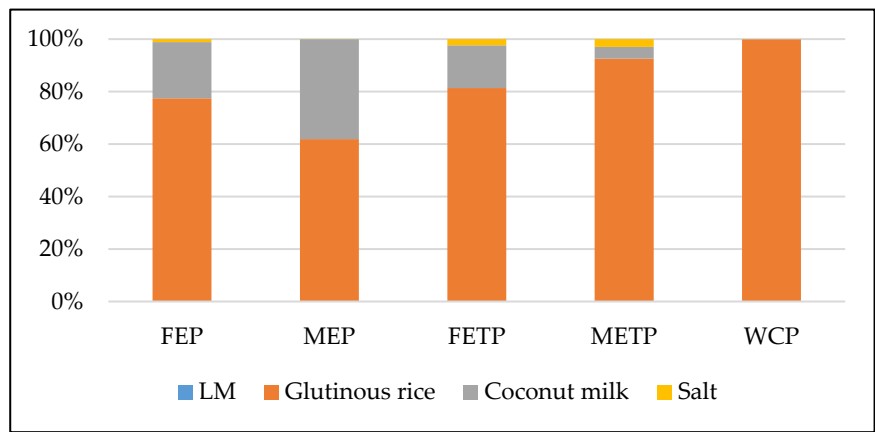

**Figure 4.** Relative contribution of lemang (LM) production to freshwater eutrophication (FEP), marine eutrophication (MEP), freshwater ecotoxicity (FETP), marine ecotoxicity (METP) and water consumption (WCP).

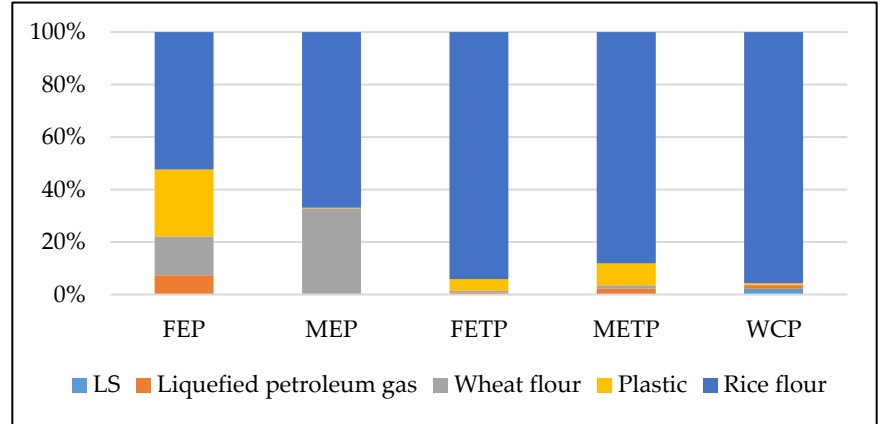

**Figure 5.** Relative contribution of noodle laksam (LS) production to freshwater eutrophication (FEP), marine eutrophication (MEP), freshwater ecotoxicity (FETP), marine ecotoxicity (METP) and water consumption (WCP).

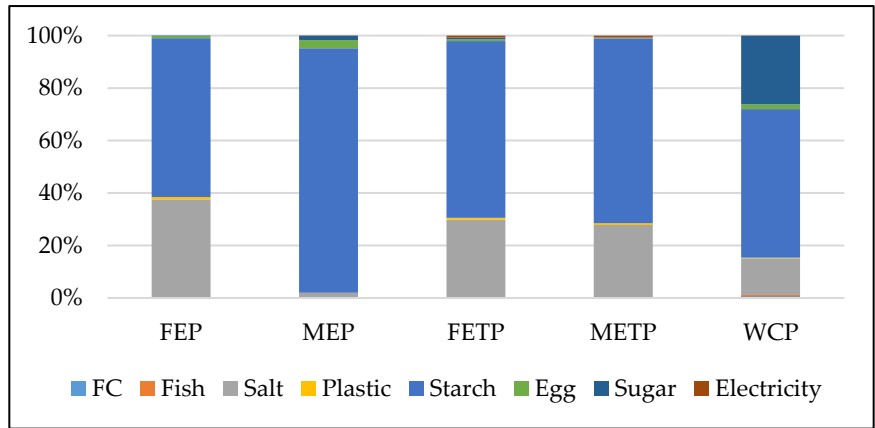

**Figure 6.** Relative contribution of fish cracker (FC) production to freshwater eutrophication (FEP), marine eutrophication (MEP), freshwater ecotoxicity (FETP), marine ecotoxicity (METP) and water consumption (WCP).

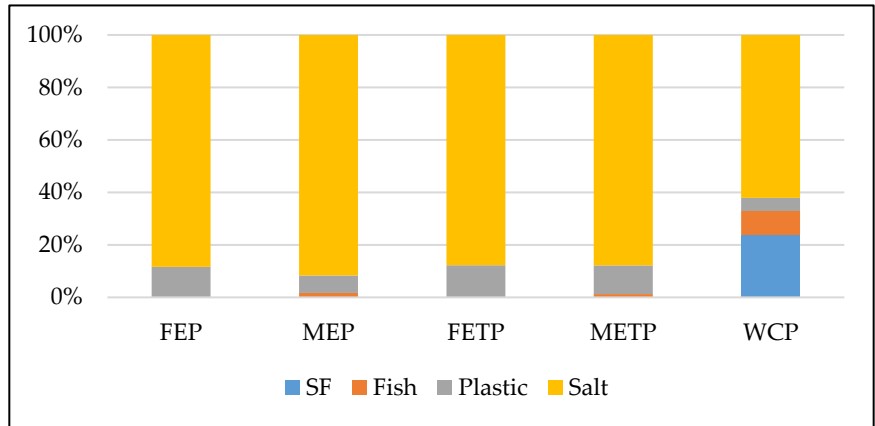

**Figure 7.** Relative contribution of salted fish (SF) production to freshwater eutrophication (FEP), marine eutrophication (MEP), freshwater ecotoxicity (FETP), marine ecotoxicity (METP) and water consumption (WCP).

### 3.3.1. Freshwater Eutrophication (FEP)

Figure 8 shows the impact at the selected food types on FEP. Freshwater eutrophication (FEP) was highest in FC and LM production and lowest in LS production. Production of FC and LM were found to be the main contributors for FEP with both at 0.572 kg P eq. From FC production, the significant contributions to FEP was from the production of starch and salt at 0.346 kg P eq and 0.215 kg P eq respectively. Electricity consumed at the factory was found to be the least significant at $2.1 \times 10^{-8}$ kg P eq. Based on Ecoinvent 3.4 Database [25] and Agri-footprint 4.0 Database [40], it was found that Phosphate ($PO_4^{3-}$) emission was 0.537 kg P eq. The major ingredient of crisps, which is a similar product to FC, and FEP were also emissions mainly from agricultural fertilizer residues (various phosphorus compounds), and a relatively small contribution was from factory activity [37].

For LM production, significant contributions to FEP were from glutinous rice and coconut milk production at 0.443 kg P eq and 0.123 kg P eq respectively. Salt consumed was the least significant at 0.006 kg P eq. Based on Ecoinvent 3.4 Database [25] and Agri-footprint 4.0 Database [40], it was found that $PO_4^{3-}$ and Phosphorus (P) emissions were 0.323 kg P eq and 0.247 kg P eq respectively. $PO_4^{3-}$ was emitted from spoils of lignite and hard coal mining at 0.118 kg P eq and 0.1 kg P eq respectively. P was mainly emitted from production of glutinous rice and coconut at 0.126 kg P eq and 0.121 kg P eq respectively. The major cause of LM production to FEP is in line with the similar product of rice stated by Jimmy et al. [34].

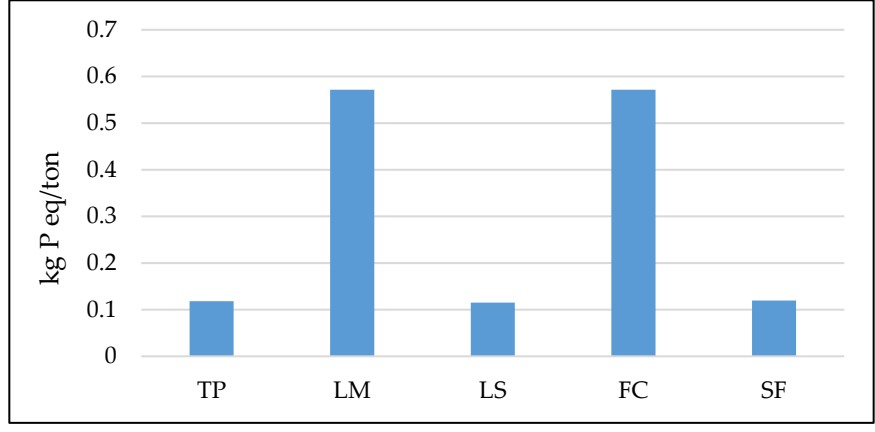

**Figure 8.** Process contribution from selected food products to freshwater eutrophication (FEP). TP is tempe, LM is lemang, LS is noodle laksam, FC is fish cracker and SF is salted fish.

LS production was found to be the smallest contributor to FEP at 0.115 kg P eq. The production of rice flour and plastic were found to be the most significant contributions to FEP at 0.077 kg P eq and 0.039 kg P eq respectively. Liquefied petroleum gas consumed was found to be the least significant at 0.008 kg P eq. Based on Ecoinvent 3.4 Database [25] and Agri-footprint 4.0 Database [40], GHG emissions were P and $PO_4^{3-}$ at 0.076 kg P eq and 0.038 kg P eq respectively. P was found to be emitted from the production of rice at 0.057 kg P eq. $PO_4^{3-}$ was mainly emitted from spoils of lignite and hard coal mining at 0.021 kg P eq and 0.014 kg P eq respectively.

### 3.3.2. Marine Eutrophication (MEP)

Marine eutrophication (MEP) is the second type of damage impact category analyzed. Figure 9 summarizes the impacts of producing the selected food types on MEP. The most significant impact on MEP was LM and FC production at 4.281 kg N eq and 2.231 kg N eq, respectively. Based on site investigation in this study, 833.333 kg of glutinous rice and 8333.333 kg of coconut milk were required to produce 1 ton of LM, which produced MEP of 2.646 kg N eq and 1.631 kg N eq, respectively. In addition, 1100 kg of starch and 100 eggs were required to produce 1 ton of FC. These raw materials produced 2.078 kg N eq and 0.069 kg N eq.

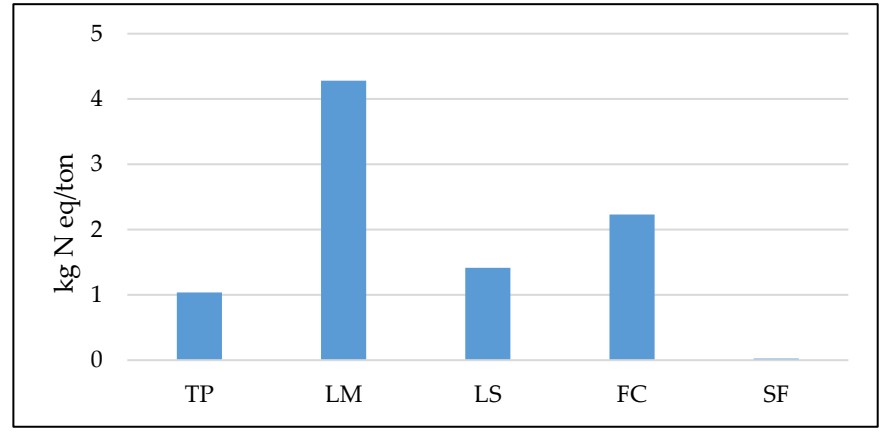

**Figure 9.** Process contribution from selected food products to marine eutrophication (MEP). TP is tempe, LM is lemang, LS is noodle laksam, FC is fish cracker and SF is salted fish.

Based on Ecoinvent 3.4 Database [25] and Agri-footprint 4.0 Database [40], emissions from LM and FC production were Nitrate ($NO_3^-$) at 4.28 kg N eq and 2.21 kg N eq, respectively. Nitrate was emitted from the production process of rice to produce glutinous rice, coconut production to produce coconut milk, and potato production process to produce starch. The latter finding of LM production is in line with the MEP of a similar product of rice described by Jimmy et al. [34] with a major cause of MEP being emission from paddy fields.

In this study, it was found that the least significant contributors for MEP was SF production at 0.024 kg N eq. Salt production was the most significant contribution at 0.022 kg N eq. On the other hand, SF production process in the factory was found to be zero contribution to MEP.

### 3.3.3. Freshwater Ecotoxicity (FETP)

Figure 10 summarizes the contributions from selected food products to FETP. Production of LS had the highest FETP at 74.075 kg 1, 4-DCB. The main contribution was consumption of rice flour at 69.701 kg 1, 4-DCB. The results show that cultivation of rice in intensive fields was the major cause of impact, which is the same result as the past study on LCA of pasta production—a similar product to LS—which also showed that the cultivation of wheat in intensive fields was the main contributor of FETP [38]. Based on Ecoinvent 3.4 Database [25] and Agri-footprint 4.0 Database [40] adopted in this study, the emissions during the production of these selected food types was Cypermethrin

($C_{22}H_{19}Cl_2NO_3$) at 40.4 kg 1, 4-DCB. $C_{22}H_{19}Cl_2NO_3$ was emitted from the production of rice flour used at 40.4 kg 1, 4-DCB.

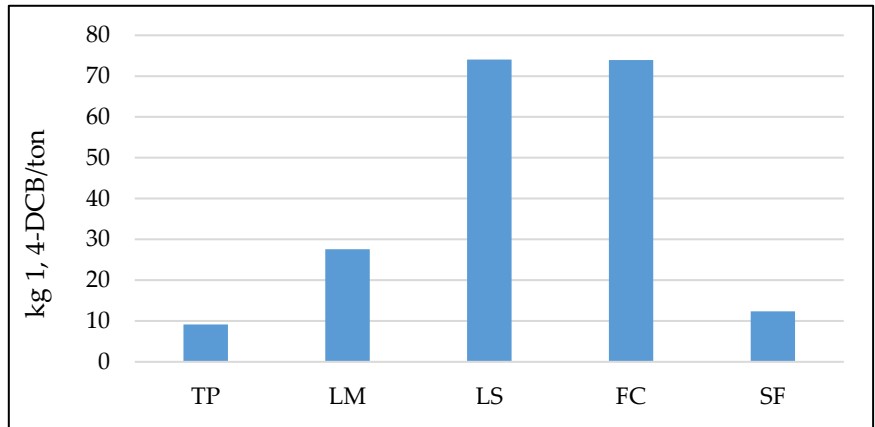

**Figure 10.** Process contribution from selected food products to freshwater ecotoxicity (FETP). TP is tempe, LM is lemang, LS is noodle laksam, FC is fish cracker and SF is salted fish.

The production of TP was the least significant contributor for FETP at 9.166 kg 1, 4-DCB. The production of soybeans was found to be the most significant contribution to FETP at 8.566 kg 1, 4-DCB. Based on site investigations, it was found that the process line to produce TP in the factory was zero FETP. The product emitted from the process to produce soybean was Diflubenzuron ($C_{14}H_9ClF_2N_2O_2$) at 7.357 kg 1, 4-DCB.

3.3.4. Marine Ecotocixity (METP)

METP is the fourth type of damage impact category analyzed. The total characterization factor for METP was found at 216.218 kg 1, 4-DCB. Figure 11 summarizes the contributions resulted from the selected food products to METP. The most significant impact to METP was found to be the production of FC and LS at 113.4 kg 1, 4-DCB and 49.16 kg 1, 4-DCB respectively. The main contributions from FC were the production of starch and salt at 79.9 kg 1, 4-DCB and 31.455 kg 1, 4-DCB respectively. The production of rice flour and plastic for LS production were found to emit 43.322 kg 1, 4-DCB and 4.125 kg 1, 4-DCB respectively. The latter findings from FC production are in line with the ecotoxicity impacts of a similar product of crisps described by Lansche et al. [30].

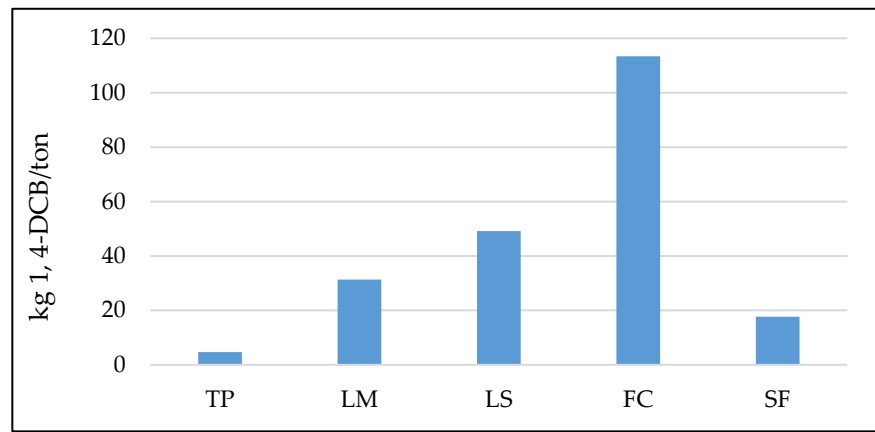

**Figure 11.** Process contribution from selected food products to marine ecotoxicity (METP). TP is tempe, LM is lemang, LS is noodle laksam, FC is fish cracker and SF is salted fish.

Based on Ecoinvent 3.4 Database [25] and Agri-footprint 4.0 Database [40], emissions from FC production was Zn at 60.263 kg 1, 4-DCB while LS production was $C_{22}H_{19}Cl_2NO_3$ at 34.62 kg 1, 4-DCB. Zinc was emitted from the sulfidic tailing process to produce starch and salt for FC production at 26.8 kg 1, 4-DCB and 20.7 kg 1, 4-DCB respectively. Other than that, $C_{22}H_{19}Cl_2NO_3$ was found to be emitted from the production of rice flour for LS production at 34.62 kg 1, 4-DCB.

In this study, it was found that the least significant contributors for METP was TP production at 9.166 kg 1, 4-DCB of which soybean production accounted for 4.67 kg 1, 4-DCB. Tempe production process in the factory was found not to contribute to METP. Based on Ecoinvent 3.4 Database [25] and Agri-footprint 4.0 Database [40], GHG emissions from TP production were $C_{14}H_9ClF_2N_2O_2$ and Zn at 0.935 kg P eq and 0.531 kg P eq respectively. It was found that $C_{14}H_9ClF_2N_2O_2$ was emitted during the production of soybeans at 1.442 kg 1, 4-DCB. Zinc at 1.3 kg 1, 4-DCB was produced mainly from sulfidic tailing process for soybean production.

### 3.3.5. Water Consumption (WCP)

Figure 12 shows the impacts of selected food types on WCP. The total characterization factor was for WCP was found to be at 654.99 $m^3$. Based on Hanafiah et al. (2011) [44], Huijbregts et al. (2016) [41] and Pfister et al. (2009) [45], the high consumption of water could lead to a rise in malnutrition as food production requires water. It could also cause an impact on freshwater and terrestrial species due to lack of freshwater.

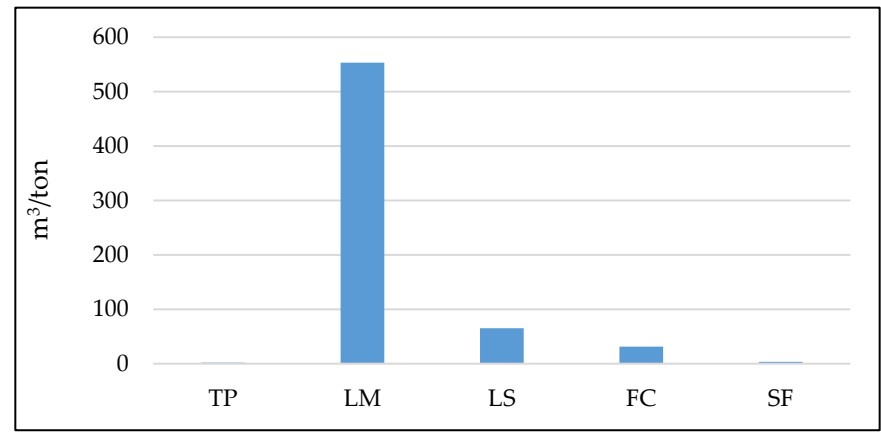

**Figure 12.** Process contribution from selected food products to water consumption (WCP). TP is tempe, LM is lemang, LS is noodle laksam, FC is fish cracker and SF is salted fish.

Lemang production had the highest impact on WCP at 553.078 $m^3$. The main contribution from LM production to WCP was rice irrigation at 522.427 $m^3$. For LM factory production and salt were the least significant at 0.081 $m^3$ and 0.13 $m^3$. The smallest contributor to WCP was the production of TP at 2.0145 $m^3$. Production of TP was the main contribution to WCP at 2.01 $m^3$. Electricity production had the lowest WCP at $1.75 \times 10^{-5}$ $m^3$.

## 4. Conclusions

In conclusion, this is the first study presenting the findings from WFA and LCA on five traditional food productions in Malaysia, a study which previously did not exist in the literature. The WFA of selected traditional food products, TP, LM, LS, FC and SF were established in this study. Assessments of environmental impacts resulting from the food production based on LCA framework and methodology were presented.

It was found that consumption of water in LM production was higher than other selected traditional food productions. Total WF of LM production was found to be 3862.133 $m^3$/ton, while the lowest total WF was the production of FC, at 135.88 $m^3$/ton. The study indicates that the biggest

contribution to the volumetric WF of LM is from coconut grown with green water. Results from this study showed that the $WF_{green}$ on all selected food products is higher than its $WF_{blue}$ and $WF_{grey}$. Water withdrawal activities at all the watersheds of the study areas are still acceptable since the ratio of water withdrawal to hydrological availability is found to be at low stress level. The results on $WS_{blue}$ and WPL obtained from this study, show that both values did not exceed 1%, therefore it can be concluded that all the selected food products in this study can be considered sustainable.

An LCA was conducted to assess the environmental performance of the selected traditional food products based on the water consumption and effluent emissions. The cradle-to-gate approach adopted in this study includes the raw materials used and the processing line. Effluent emissions were discussed in terms of its potential environmental impacts. It was found that the process to produce FC was the main contributor to FEP and METP due to the production of starch, which is used to produce FC. However, the production of LM was significant for MEP and WCP. The findings found that the cultivation of glutinous rice from paddy field and irrigation of rice were the major causes of MEP and WCP, while LS production was the main contributor to FETP due to the emissions from cultivation of rice. As stated in the results, the production of raw materials used is the factor that has significant impact on the environment. Having said that, the production process of selected food products in the factory creates less water-related impact compared to the production of raw materials.

*Recommendations for Future Studies*

Based on findings established in this study, relevant organizations and authorities engaged in the food industry need to further improve their water efficiency and consumption. It is proposed that food production strategy, and in particular the guidelines on agriculture and food manufacturing activities, be revisited. Suitable metrics to manage the sustainable use of water resources for water stress reduction need to be established owing to the increasing demand of industrial practices, particularly in food manufacturing.

It is suggested that in the future a comprehensive study will need to be conducted to assess the impacts of nutrient enrichment on grey water resulting from food production factories. Similarly, other food types consumed in Malaysia can further be investigated. The water scarcity index and ReCiPe 2016 derived in this study can serve as characterization factors in LCA at midpoint level. It is recommended that LCA at endpoint level be conducted to further assess the impacts of water consumption to human health and ecosystems.

**Author Contributions:** P.X.H.B. was involved in conducting the study on-site with assistance from M.A.M., N.H.M., and M.M.H. by giving advice and recommendation to further improved the research. P.X.H.B., M.A.M., N.H.M., and M.M.H. were actively involved in drafting the manuscript. All authors have read and agreed to the published version of the manuscript.

**Funding:** This research was funded by UNITEN RMC Internal Research Grant, grant number [RJO 10517919/iRMC/Publication].

**Acknowledgments:** The authors would like to acknowledge technical and financial support from UNITEN RMC Internal Research Grant: RJO 10517919/iRMC/Publication and Drainage Irrigation Department (JPS) Malaysia. The author would also like to acknowledge Institute of Sustainable Energy, Universiti Tenaga Nasional Malaysia, and Faculty of Science and Technology, Universiti Kebangsaan Malaysia for technical and financial support.

**Conflicts of Interest:** The authors declare no conflicts of interest.

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
