# Peer review of "Cradle-to-Gate Water-Related Impacts on Production of Traditional Food Products in Malaysia"

_sustainability, doi:10.3390/su12135274_

Round 1

Reviewer 1 Report

it is advisable to make accurate comparisons with similar analyzes using LCA applied to processes with traditional culture and with other types of food at an international level. 
in references, it is recommended to include:

Ingrao C., Selvaggi R., Valentia F., Matarazzo A., Pecorino B., Arcidiacono C. (2019), Life cycle assessment of expanded clay granulate production using different fuels. RESOURCES CONSERVATION AND RECYCLING, vol. 141, p. 398-409, ISSN: 0921-3449, doi: 10.1016/j.resconrec.2018.10.026;

Ingrao C., Matarazzo A., Tricase C., Clasadonte M T, Huisingh D. (2015). Life Cycle Assessment for Highlighting Environmental Hotspots in Sicilian Peach Production Systems. JOURNAL OF CLEANER PRODUCTION, vol. 92, p. 109-120, ISSN: 0959-6526;

Author Response

Point 1: It is advisable to make accurate comparisons with similar analyzes using LCA applied to processes with traditional culture and with other types of food at an international level. 

Response 1: The comparisons were added.

Point 2: In references, it is recommended to include:

Ingrao C., Selvaggi R., Valentia F., Matarazzo A., Pecorino B., Arcidiacono C. (2019), Life cycle assessment of expanded clay granulate production using different fuels. RESOURCES CONSERVATION AND RECYCLING, vol. 141, p. 398-409, ISSN: 0921-3449, doi: 10.1016/j.resconrec.2018.10.026;

Ingrao C., Matarazzo A., Tricase C., Clasadonte M T, Huisingh D. (2015). Life Cycle Assessment for Highlighting Environmental Hotspots in Sicilian Peach Production Systems. JOURNAL OF CLEANER PRODUCTION, vol. 92, p. 109-120, ISSN: 0959-6526

Response 2: All the references were added.

Reviewer 2 Report

I have made extensive grammar and scientific corrections.  I understand that you are not native English speakers, but your message often times gets lost trying to understand what you have written.  In the future, it would best for you to have a native English speaker edit your papers before submitting, as there were numerous grammatical mistakes.  There are also some areas where you need to clarify your meaning.  In general, your study was sound and you have good information to present, you just need to get it in a format that is easier for your reader to understand.  I made some recommendations regarding adding tables and some other papers that you can read/cite for reference.  Focus on the most important aspects of what you found instead of saying all of the findings, and refer the reader to tables and figures for additional details.  You also need to rewrite all figure and table captions to convey the important information for your study in each one.

Author Response

Point 1: I have made extensive grammar and scientific corrections.

Response 1: The corrections have been made.

Point 2: There are also some areas where you need to clarify your meaning.

Response 2: The meaning of the areas have been clarified.

Point 3:  I made some recommendations regarding adding tables and some other papers that you can read/cite for reference.

Response 3: The recommendations have been added.

Point 4: Focus on the most important aspects of what you found instead of saying all of the findings, and refer the reader to tables and figures for additional details.

Response 4: The important aspects have been improved.

Point 5: You also need to rewrite all figure and table captions to convey the important information for your study in each one.

Response 5: The figure and table captions have been rewrite.

Reviewer 3 Report

Dear Authors,

The water footprint and water related impacts were analysed for five typical meals in Malaysia. Data from manufacturing was collected on-site, which is valuable. It is nice to see more research on food from countries like Malaysia, with a vast cuisine. The manuscript is very interesting, however there are several issues that need to be improved. Please, see below detailed comments and suggestions per section:

Title 

I think you could modify the title explaining that the analysis only includes water related impacts. I also wonder why only this if you have a full set of impacts to analyse and make this research more robust.

Introduction

It seems odd to read about European examples in paragraph (line 42-46) and not see any example of Malaysian Food or at least Regional Food. There are strong research groups working on food systems in the region, which I guess would be good to acknowledge together with the effort mentioned by Malaysian researchers (paragraph line 66-74).

Paragraph line 56-65 should be moved to methodology.

It might be good to develop more the product selection process explaining why these products are so important. Please consider adding context in terms of local consumption (e.g. weekly/monthly consumption, tonne production, or similar). 

Methodology

The structure of this section needs to be analysed. If this study follows the ISO standard 14040/44, there is a specific structure that needs to be follow which here is not seen. Please modify accordingly. For example, the goal and scope was defined in the introduction, but not in the methodology.

I suggest the authors to be clearer in the inventory section. Add heading to explain products and its compositions (e.g. ingredients quantity, etc.). Similarly, the explanation of every stage is missing (e.g. what is accounted for, using what database, etc). The inventory section is not clear and making difficult the potential replication of the study. 

Assumptions are not clear. Line said that green water is not accounted but then it is explained that is important for raw materials.

The functional unit is not adequate. If the study assessed the water footprint and some water related impact under a life cycle assessment methodology, the F.U. does not reflect the actual products, which I guess are solid and sell/produce per kg/package, etc.

There is semantic problem. Midpoints is refer as the approach but then the impact as refer as damage, which refers to end-point.

I suggest join table 1-5 together, to safe space and enable comparison. 

Inventory data and explanation of stages and assumptions are missing (e.g. specific datasets, what is accounted and what not, description of what each stage includes, etc.). What about waste across the supply chain and production? what about waste management? transport? etc.

Results

The presentation of the environmental impacts and water footprint needs to be improved. It is important to understand the contribution of each life cycle stage, that is not shown. You could add this in figure 4-8, develop a new figure from figure 2. You could put figure 4-8 in 2 plots instead of single plots.

It is not clear why would you show the impacts as Fig 3. Are these products from the same company?  did you put volume of sales to understand actual impact? for comparison, you already have figure 4.

Why didn't you analyse more impacts if you already have all?

What about validation of results? sensitivity analysis?

Conclusion:

It talks about a comprehensive review that wasn't found. 

Author Response

Point 1: I think you could modify the title explaining that the analysis only includes water related impacts. I also wonder why only this if you have a full set of impacts to analyse and make this research more robust.

Response 1: The title have been changed. My scope of study is looked at the water related impacts only.

Point 2: It seems odd to read about European examples in paragraph (line 42-46) and not see any example of Malaysian Food or at least Regional Food. There are strong research groups working on food systems in the region, which I guess would be good to acknowledge together with the effort mentioned by Malaysian researchers (paragraph line 66-74).

Response 2: There is no water footprint of food products had been done in Malaysia yet. So I have been added the example of Malaysia's growing oil palm. 

Point 3: Paragraph line 56-65 should be moved to methodology.

Response 3: Paragraph lines 56-65 have been moved to methodology.

Point 4: It might be good to develop more the product selection process explaining why these products are so important. Please consider adding context in terms of local consumption (e.g. weekly/monthly consumption, tonne production, or similar). 

Response 4: The reason of selecting these products has been mentioned in system boundary. The context in terms of local consumption was added too.

Point 5: The structure of this section needs to be analysed. If this study follows the ISO standard 14040/44, there is a specific structure that needs to be follow which here is not seen. Please modify accordingly. For example, the goal and scope was defined in the introduction, but not in the methodology.

Response 5: The structure has been analysed. Goal and scope definition and life cycle inventory were written in Methodology. Life cycle analysis results and interpretation were written in Results.

Point 6: I suggest the authors to be clearer in the inventory section. Add heading to explain products and its compositions (e.g. ingredients quantity, etc.). Similarly, the explanation of every stage is missing (e.g. what is accounted for, using what database, etc). The inventory section is not clear and making difficult the potential replication of the study.

Response 6:  The inventory section has been improved.

Point 7: Assumptions are not clear. Line said that green water is not accounted but then it is explained that is important for raw materials.

Response 7: The explanation has been added in paragraph line 134-138..

Point 8: The functional unit is not adequate. If the study assessed the water footprint and some water related impact under a life cycle assessment methodology, the F.U. does not reflect the actual products, which I guess are solid and sell/produce per kg/package, etc.

Response 8: The functional unit is per ton of product. I already added for water footprint and water related impacts.

Point 9: There is semantic problem. Midpoints is refer as the approach but then the impact as refer as damage, which refers to end-point.

Response 9: The problem has been explained.

Point 10: I suggest join table 1-5 together, to safe space and enable comparison. 

Response 10: Table 1-5 have been combined. 

Point 11: Inventory data and explanation of stages and assumptions are missing (e.g. specific datasets, what is accounted and what not, description of what each stage includes, etc.). What about waste across the supply chain and production? what about waste management? transport? etc.

Response 11: Inventory data and explanation have been added.

Point 12: The presentation of the environmental impacts and water footprint needs to be improved. It is important to understand the contribution of each life cycle stage, that is not shown. You could add this in figure 4-8, develop a new figure from figure 2. You could put figure 4-8 in 2 plots instead of single plots.

Response 12: The presentation of contribution on each food type have been shown in Fig 3-7. 

Point 13: It is not clear why would you show the impacts as Fig 3. Are these products from the same company?  did you put volume of sales to understand actual impact? for comparison, you already have figure 4.

Response 13: Fig 3 has been separated to Fig 3-7. Figure 8-12 are used to compare.

Point 14: Why didn't you analyse more impacts if you already have all?

Response 14: My data is not enough to assess for all the impacts. My scope of study is looked at the water related impacts only.

Point 15: Conclusion: It talks about a comprehensive review that wasn't found. 

Response 15: Conclusion has been improved.

Round 2

Reviewer 1 Report

improve conclusion and references; improve comparisons with similar analyzes using LCA with tables

Author Response

Point 1: Improve conclusion and references.

Response 1: Conclusion and references were improved.

Point 2: Improve comparisons with similar analyzes using LCA with tables.

Response 2: Similar analysis using LCA with table were added.

Reviewer 2 Report

The authors made major improvements to the article.  There are still a few minor corrections mainly for grammar and readability.  The authors should also  expand the figure and table legends to be more comprehensive.

Author Response

Point 1: The authors made major improvements to the article.  There are still a few minor corrections mainly for grammar and readability.  

Response 1: The corrections have been made based on the provided file.

Point 2: The authors should also  expand the figure and table legends to be more comprehensive.

Response 2: The figure and table legends have been expand.